# The Double-Ellipsoid Geometry of CLIP

**Meir Yossef Levi** [1]   **Guy Gilboa** [1]

## Abstract

Contrastive Language-Image Pre-Training (CLIP) is highly instrumental in machine learning applications within a large variety of domains. We investigate the geometry of this embedding, which is still not well understood, and show that text and image reside on linearly separable ellipsoid shells, not centered at the origin. We explain the benefits of having this structure, allowing to better embed instances according to their uncertainty during contrastive training. Frequent concepts in the dataset yield more false negatives, inducing greater uncertainty. A new notion of conformity is introduced, which measures the average cosine similarity of an instance to any other instance within a representative data set. We prove this measure can be accurately estimated by simply computing the cosine similarity to the modality mean vector. Furthermore, we find that CLIP's modality gap optimizes the matching of the conformity distributions of image and text.

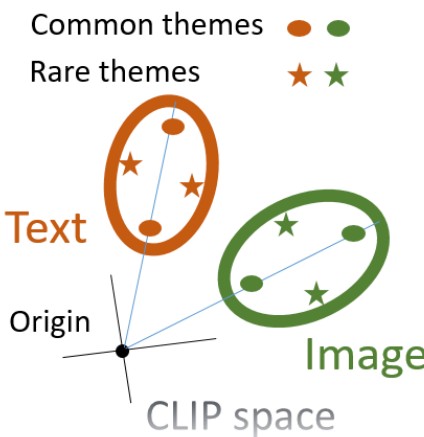

Figure 1. Sketch of CLIP general geometry: image and text are embedded on linearly separable ellipsoid shells, not centered at the origin. This allows to control uncertainty in contrastive learning, where as themes become more rare (lower uncertainty) they reside farther from the mean modality vector.

## 1. Introduction

Multi-modal approaches, particularly Contrastive Language-Image Pre-Training (CLIP) (Radford et al., 2021), have revolutionized computer vision tasks, enabling applications such as high-quality image generation (Ramesh et al., 2022; Nichol et al., 2021), open-vocabulary classification (He et al., 2023), segmentation (Liang et al., 2023; Yu et al., 2024), detection (Wu et al., 2023), captioning (Mokady et al., 2021; Cho et al., 2022), and semantic editing (Kim et al., 2022; Kawar et al., 2023). Beyond images, CLIP's success extends to 3D (Hegde et al., 2023; Chen et al., 2023; Zhang et al., 2022), video (Tang et al., 2021; Luo et al., 2022), and audio domains (Wu et al., 2022; Guzhov et al., 2022).

Despite these advances, the structure of CLIP's latent space remains poorly understood. Existing studies focus on properties like alignment, uniformity, and the modality gap (Liang et al., 2022) but overlook the geometry underlying this multi-modal space. The L2-normalization phase, which is integral when performing cosine similarity, practically reducing the data to the unit hypersphere. Since normalization is an information-reducing process, understanding the primary embeddings prior to normalization can reveal deeper insights into the latent space geometry.

In this paper, we propose analyzing the pre-normalized CLIP primary embedding for three key reasons: (1) **Enhancing downstream tasks.** While $L^2$-normalization is integral to the cosine similarity used during training, the primary embedding is directly employed in critical downstream tasks, including image generation and semantic editing. Analysis of the latent geometry can enhance the performance of these tasks. (2) **Semantic significance of magnitude.** Despite the cosine similarity is agnostic to the norm, we observe that magnitude still plays a significant and meaningful role. Notably, the largest embeddings in MS-COCO correspond

[1]Viterbi Faculty of Electrical and Computer Engineering, Technion - Israel Institute of Technology, Haifa, Israel. Correspondence to: Meir Yossef Levi <me.levi@campus.technion.ac.il>, Guy Gilboa <guy.gilboa@ee.technion.ac.il>.

*Proceedings of the 42^{nd} International Conference on Machine Learning*, Vancouver, Canada. PMLR 267, 2025. Copyright 2025 by the author(s).

to unusual or exotic captions (e.g., "I am not sure what this image is", see full histogram and examples in Figure 14 in the Appendix). (3) **Deeper understanding of contrastive learning.** CLIP is an exceptional semantic encoder achieved through a rather generic contrastive loss and huge training data. Investigating the solutions found by CLIP allows deeper insights on contrastive learning, possible approaches to tackle false negatives and may shed light on unresolved phenomena, such as *the modality* gap and *the narrow cone effect* (Liang et al., 2022).

Our analysis reveals that CLIP's primary latent space exhibits a double-ellipsoid geometry, with one ellipsoid for images and another for text. Both are shifted from the origin (see Fig. 1), in line with the *narrow cone effect* and the *modality gap* (Liang et al., 2022; Fahim et al., 2024; Schrodi et al., 2024). Using the MS-COCO validation set (Lin et al., 2014), we show that both modalities exhibit the thin-shell phenomenon (Klartag, 2023; Klartag & Lehec, 2022), where most of the mass concentrates within a specific range from the mean.

This geometry affords several advantages. The offset from the origin allows CLIP to control the sharpness of its response in contrastive learning, mitigating false negatives (Byun et al., 2022; Li et al., 2022; Yang et al., 2022); instances that are conceptually similar but incorrectly treated as negatives. Frequent concepts with higher uncertainty are embedded closer to the mean vector, a phenomenon we term *semantic blurring*, reducing loss and improving performance. Our experiments confirm that frequent concepts are better aligned to the mean vector of the ellipsoid, achieving excellent agreement with our hypothesis.

Leveraging this deeper understanding, we introduce a new definition of concept *conformity*, quantifying how close a sample resides with respect to all others. We prove that there is a proportion between conformity and cosine similarity to the mean vector (See proof in Supp. C1, and empirically with Pearson correlation: 0.9998 for MS-COCO). Furthermore, we show that the distribution of conformity differs between modalities, with CLIP's ellipsoid alignment offering a plausible explanation for the modality gap.

Our contributions are as follows:

1. We reveal that CLIP embeddings form separable ellipsoid shells for each modality, shifted from the origin.

2. We analyze the benefits of this structure, including its role in controlling sharpness in contrastive learning.

3. We show that frequent concepts benefit most from this geometry, optimizing the contrastive loss near the ellipsoid offsets for MS-COCO.

4. We define concept *conformity* and demonstrate its

strong correlation with similarity to the mean vector, offering insights into semantic organization.

5. We highlight the role of conformity in explaining the modality gap and propose its use in ranking text and image generators.

6. We introduce *vertical SLERP (vSLERP)*, an interpolation method leveraging the geometry of CLIP's latent space.

## 2. Related Work

Contrastive representation learning is a powerful learning scheme, where models are trained to associate positive pairs (e.g., different views of the same image (Chen et al., 2020)) closely in the embedding space while pushing negative pairs (e.g., different images) apart. This simple yet effective approach has led to significant advances across a wide range of applications, i.e. image classification (Chen et al., 2020; He et al., 2020), natural language processing (Gao et al., 2021; Kim et al., 2021), 3D analysis (Afham et al., 2022; Xie et al., 2020) and more.

The latent space induced by contrastive learning has been widely explored (Arora et al., 2019; Ji et al., 2023; Wang et al., 2022; Wang & Isola, 2020), often conceptualized as a normalized hypersphere (Wang & Isola, 2020; Liang et al., 2022). Alignment and uniformity (Wang & Isola, 2020) are key properties of the *Normalized Temperature-scaled Cross-Entropy (NT-Xent)* loss (Chen et al., 2020). Optimizing alignment and uniformity was shown to be crucial for preserving rich semantic structures in the latent space, leading to improvements in downstream performance across multiple domains (Fahim et al., 2024).

With the rise of cross-modal contrastive models, such as CLIP (Radford et al., 2021), which align images and text in a shared embedding space, new challenges in latent space geometry have emerged. A notable issue is the modality gap (Liang et al., 2022), where embeddings from different modalities, such as images and text, are separated in the shared latent space. Moreover, the narrow cone effect was observed (Liang et al., 2022; Schrodi et al., 2024), where features occupy only a limited portion of the angular space.

One of the main challenges in multimodal contrastive learning is of obtaining high-quality pairs. Web-scale datasets may include mismatched positive pairs (Chun et al., 2022; Gadre et al., 2024; Maini et al., 2023; Wang et al., 2023) or mislabeled negative pairs that are actually positive, referred to as *false negatives* (Byun et al., 2022; Li et al., 2022; Yang et al., 2022). Numerous approaches have emerged to address this challenge, such as by identifying and introducing hard negative examples (Byun et al., 2024; Chuang et al., 2020; Robinson et al., 2020; Kalantidis et al., 2020). Our obser-

vations are that false negatives appear to play a significant role in forming the geometry of CLIP's latent space.

## 3. Random vectors in high dimensions

### 3.1. Notations

We investigate CLIP space induced by ViT-B/32 encoders of $n = 512$ dimensions, $\mathcal{X} = R^n$. Let $\mathcal{X}_i \subset \mathcal{X}$ be the *image* subspace and $\mathcal{X}_t \subset \mathcal{X}$ be the *text* (captions) subspace. We will reaffirm that they are different and in fact linearly separable (Schrodi et al., 2024; Liang et al., 2022). Let $v \in \mathcal{X}$ be a vector in this space. We denote by $v_i \in \mathcal{X}_i$ vectors of images and by $v_t \in \mathcal{X}_t$ vectors of text. The symbol $\mathbb{E}$ stands for the expected value. The respective modality mean of image and text are $m_i = \mathbb{E}_{v_i \in \mathcal{X}_i}[v_i]$ and $m_t = \mathbb{E}_{v_t \in \mathcal{X}_t}[v_t]$. Let $\tilde{v}$ be the vector after subtraction of the respective modality mean. That is, for images, $\tilde{v}_i = v_i - m_i : v_i \in \mathcal{X}_i$ and for text $\tilde{v}_t = v_t - m_t : v_t \in \mathcal{X}_t$.

Our statistical analysis and many experimental results are based on MS-COCO (Lin et al., 2014) validation set, a common standard image-text dataset.

### 3.2. High dimensional geometry of random vectors

It is often challenging to obtain good intuition on the probability manifold and its geometry in high dimensions. We outline below some fundamental concepts.

#### 3.2.1. THIN SHELL THEORY

There is an intensive research related to the thin shell phenomenon (Kannan et al., 1995; Paouris, 2006; Klartag & Lehec, 2022; Jambulapati et al., 2022; Klartag, 2023). Definitions of *log concave distributions* and *isotropic random vectors* appear in the Appendix. Since isotropic random vectors have a unit second moment for any $x(k)$, $k = 1, ..n$, we get that the expected value of the squared Euclidean norm is

$$\mathbb{E}[\|x\|^2] = \mathbb{E}[\sum_{k=1}^{n} x(k)^2] = \sum_{k=1}^{n} \mathbb{E}[x(k)^2] = n. \quad (1)$$

As shown for example in (Paouris, 2006), $\mathbb{E}[\|x\|^2] \approx \mathbb{E}^2[\|x\|]$, the expected norm of $x$ can be approximated by

$$\mathbb{E}[\|x\|] \approx \sqrt{n}. \quad (2)$$

For isotropic log-concave distributions we have the *thin shell* property:

**Theorem 3.1** (Thin shell). *Let the thin shell parameter be defined by*

$$\sigma_n^2 = \sup_x \mathbb{E}(\|x\| - \sqrt{n})^2,$$

*where the supremum is over isotropic, log-concave random vectors in $R^n$. Then $\sigma_n \leq c(\log n)^\alpha$, where $c$ is a universal constant.*

Recent studies have shown this bound for $\alpha = 4$ (Klartag & Lehec, 2022), $\alpha = 2.23$ (Jambulapati et al., 2022) and most recently for $\alpha = \frac{1}{2}$ (Klartag, 2023). See more details in the above papers and the references therein. Essentially, this means the mass of the distribution is concentrated around a shell of radius $\sqrt{n}$.

Let us farther examine this for the more general anisotropic case. Let $x = (x(1), ..., x(n))$ be a vector of $n$ random variables of different distributions (not iid), each of mean zero. Let the norm of $x$, which is a random variable, be defined by $\|x\| = \mu_{norm} + y$, where $\mu_{norm} := \mathbb{E}[\|x\|]$ and $y$ is a random variable of zero mean. We examine the term $\mathbb{E}[\|x\|^2] = tr(\mathbb{C})$, where $tr$ is the trace and $\mathbb{C}$ is the covariance matrix of $x$:

$$\begin{aligned} \mathbb{E}[\|x\|^2] &= \mathbb{E}[(\mu_{norm} + y)^2] \\ = \mathbb{E}[\mu_{norm}^2 + 2\mu_{norm}y + y^2] &= \mu_{norm}^2 + \text{var}(y). \end{aligned} \quad (3)$$

Therefore, for $\mu_{norm}^2 \gg \text{var}(y)$ we can approximate

$$\mathbb{E}[\|x\|] = \mu_{norm} \approx \sqrt{\mathbb{E}[\|x\|^2]} = \sqrt{tr(\mathbb{C})}. \quad (4)$$

Here the squared expected Euclidean norm and the trace of the covariance matrix approximately coincide. We can thus view $\text{std}(x(k))$ as a rescale of the coordinate system in dimension $k$, with respect to a unit sphere.

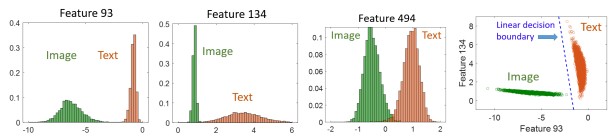

*Figure 2.* Normalized histograms of certain CLIP features. Image and text are clearly drawn from different statistics. On the right it is shown that even two features are sufficient to obtain full linear separability. The results of a linear SVM classifier are shown (blue dashed line, with $100\%$ accuracy on MS-COCO).

## 4. Geometric Analysis

We begin by examining the statistics of image and text in the CLIP embedding space $\mathcal{X}$. This part is completely data-driven without any prior assumptions related to the training process. We focus on *the primary CLIP embedding*, which is the output of the encoder before $L_2$ normalization, i.e. before projection onto the unit hypersphere. This projection loses important information. It basically "flattens" the original geometry artificially, in a manner which is hard to analyze. More details and statistical data are provided in the Appendix. Let us first examine the known modality gap phenomenon (Liang et al., 2022) in the primary embedding. In Fig. 2, normalized histograms are shown for features 93, 134 and 494 of the CLIP latent vector. We get a bimodal distribution where image and text are clearly not drawn from the same distribution. For feature 93, for instance,

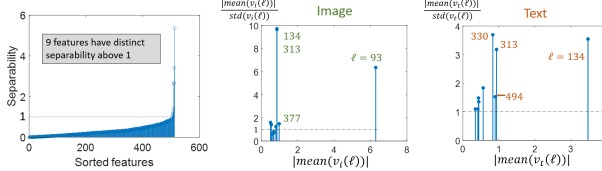

*Figure 3.* Separability of features (left) and 10 most significant features $\ell$ for image and text, with high absolute mean, compared to the feature's standard deviation.

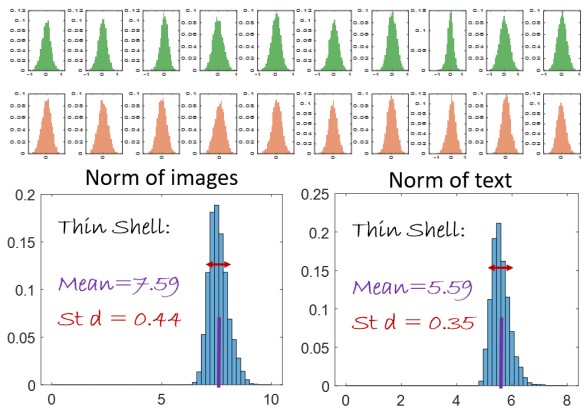

*Figure 4.* Statistics of image and text features after mean subtraction. *Top:* The first 10 features for image (top) and text (bottom). *Bottom:* Histograms of $\|\tilde{v}\|$ for images and text, showing a thin-shell phenomenon with no volume below a threshold, typical for high dimensions.

the KL-divergence between the distributions is $\approx 301$ (a value above 1 implies a considerable deviation between the distributions). It was previously shown in (Shi et al., 2023; Fahim et al., 2024; Schrodi et al., 2024) that image and text can be separated linearly. We find there are actually 9 features which serve as sort of "tags" for image and text. More formally, we can define the measure of separability of a feature $\ell$ by

$$Sep(\ell) = \frac{|m_i(\ell) - m_t(\ell)|}{\sqrt{\text{var}(v_i(\ell)) + \text{var}(v_t(\ell))}}. \tag{5}$$

A plot of the features sorted by separability is shown in Fig. 3 (left). Fig. 2 (right) shows that the modalities are linearly separable (with $100\%$ accuracy) using only two such tag features (93 and 134), based on a linear SVM classifier (decision boundary shown in blue). We can thus state the following property (which holds exactly for MS-COCO):

> **Property 1:** Image and text reside on separate subspaces, $\mathcal{X}_i \cap \mathcal{X}_t \approx \emptyset$.

In Fig. 4, we show some statistics of the features of $\tilde{v}_i$ and $\tilde{v}_t$ (where the mean is subtracted). To get impression, the first 10 features in each vector are shown for both modalities. The

distribution appears smooth, unimodal, with peak around zero. The norm $\|\tilde{v}\|$, however, is distributed within a small range (thin shell) such that there is no mass near zero.

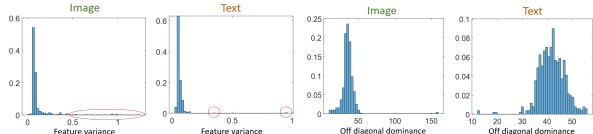

*Figure 5.* Normalized histograms of feature variance (left) show a long tail, indicating an ellipsoid rather than a hypersphere. Off-diagonal dominance (Eq. 6) suggests strong feature correlations, implying a tilted ellipsoid.

We can further check the validity of Eq. 3, we examine images here. In the case of MS-COCO statistics we have: $\mu_{norm} = 7.5873$, $\text{var}(y) = 0.1914$, yielding $\mu_{norm}^2 = 57.5671 \gg \text{var}(y)$, where the approximation $\sqrt{\mathbb{E}[\|x\|^2]} = 7.6007$ is just with $0.18\%$ relative error. We can therefore conclude:

> **Property 2:** The mass of each modality is concentrated within a thin shell, with zero mass near the mean of the distribution.

Let us now investigate the geometry of each shell. We examine the variance of each feature $\ell$. In a uniform hypersphere embedding we expect to have similar variance for all dimensions. We observe in Fig. 5 (left part) this is not the case, with a long tail distribution, where some features exhibit considerably larger variance, hence an ellipsoid structure:

> **Property 3:** The embedding of both text and image is of an ellipsoid shell.

We now examine inter-correlations between features. Let us define *off-diagonal dominance* of a row $\ell$ in the covariance matrix $\mathbb{C}$ by

$$ODD(\ell) = \frac{\sum_{k \neq \ell} |\mathbb{C}_{\ell k}|}{\mathbb{C}_{\ell \ell}}. \tag{6}$$

Diagonally dominant matrices have $ODD(\ell) < 1$, $\forall \ell$ ensuring a non-singular matrix. We observe (see Fig. 5 two right plots) that the off diagonals are significant, implying non-negligible correlation between features, thus:

> **Property 4:** The ellipsoids of both modalities are tilted.

Finally, we check the location of each ellipsoid, with respect to the origin. We recall $m_i$, $m_t \in R^n$ are the mean value vectors of image and text. Let $\sigma_i$, $\sigma_t \in R^n$ be the standard deviation vectors of image and text, respectively. We have $\frac{\|m_i\|}{\|\sigma_i\|} = 0.94$ and $\frac{\|m_t\|}{\|\sigma_t\|} = 1.03$. Viewing $\|\sigma\|$ as

a mean vector magnitude of the ellipsoid shell, the means are significantly shifted from the origin, compared to the size of the ellipsoid. This is caused by a few features, with strong deviation from the origin (compared to the respective feature's standard deviation), as shown in Fig. 3 (middle and right). Thus we can state:

> **Property 5:** The ellipsoids are not centered near the origin.

## 5. Loss behavior on a double-ellipsoid

In this section, we validate that a non-origin-centered double-ellipsoid structure achieves optimality in terms of the CLIP contrastive learning loss.

For a batch containing $M$ image-text pairs, we denote by $\bar{v}_i^j = \frac{v_i^j}{\|v_i^j\|}$ and $\bar{v}_t^j = \frac{v_t^j}{\|v_t^j\|}$ the normalized image and text features of the $j$-th pair in the batch respectively. The multi-modal learning loss used in CLIP is the *normalized temperature-scaled cross entropy loss* (NT-Xent), a variation of InfoNCE (Oord et al., 2018) loss:

$$\ell_{clip} := -\frac{1}{2} \underset{j,k \in M}{\mathbb{E}} \left[ \log \frac{e^{\bar{v}_t^{j\top} \bar{v}_i^j / \tau}}{\sum_j e^{\bar{v}_t^{j\top} \bar{v}_i^k / \tau}} + \log \frac{e^{\bar{v}_t^{j\top} \bar{v}_i^j / \tau}}{\sum_j e^{\bar{v}_t^{k\top} \bar{v}_i^j / \tau}} \right]. \tag{7}$$

As observed by (Wang & Isola, 2020), the loss can be decomposed into two terms: (1) *Alignment*, which encourages high cosine similarity for positive pairs, and (2) *Uniformity*, encourages low cosine similarity among negative ones.

$$\ell_{clip} := - \overbrace{\underset{j \in M}{\mathbb{E}} [\bar{v}_t^{j\top} \bar{v}_i^j / \tau]}^{\text{alignment}} +$$

$$\underset{k \in M}{\mathbb{E}} \left[ \overbrace{\frac{1}{2} \log \sum_{j=1}^{M} e^{\bar{v}_t^{j\top} \bar{v}_i^k / \tau} + \frac{1}{2} \log \sum_{j=1}^{M} e^{\bar{v}_t^{k\top} \bar{v}_i^j / \tau}}^{\text{uniformity}} \right]. \tag{8}$$

To empirically analyze the uniformity and alignment terms in Eq. 8 alongside the overall loss in Eq. 7, we use the MS-COCO validation set. Fig. 6 shows the overall loss (bottom) and its breakdown into uniformity and alignment losses (top). We treat the entire validation set (5k samples) as a single batch. The overall loss is further separated into correctly classified, misclassified, and combined cases; the union of the correct and misclassified is equivalent to both, and they are mutually exclusive.

In this experiment, we examine different values of the mean value of the image embedding. For simplicity, we apply

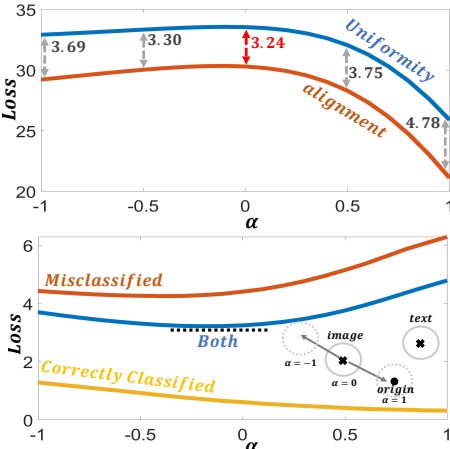

*Figure 6.* **Loss vs. embedding center position.** The parameter $\alpha$ controls the embedding center (Eq. 9, with $\alpha = 0$ as the current non-origin-centered CLIP position). *(Top).* The unified loss balances uniformity and alignment optimally for non-origin-centered positions. *(Bottom).* The loss increases for misclassified instances and decreases for well-classified ones, with balanced accuracy at $\alpha \approx 0$.

linear interpolation and extrapolation of the mean relative to the origin, using a single scalar parameter $\alpha$. This measure is conducted on a grid of $\alpha$ values from -1 to 1, with the loss calculated on image features as follows:

$$v_i^{j'} = v_i^j - \alpha \cdot m_i \quad \forall j \in M. \tag{9}$$

The values of $v_t$ remain unchanged. Unlike the *Embedding Shift Experiment* in (Liang et al., 2022), here, the modalities are not directly shifted to each other, but to the origin.

The results show that the loss for correctly classified samples decreases monotonically with the shift toward the origin (i.e. that for perfect alignment as assumed for example by (Liang et al., 2022), shifting to the origin would be preferable). Conversely, the loss for misclassified samples increases. The overall loss balances alignment and uniformity for both correctly and misclassified samples, reaching an optimal $\alpha$ near zero. This aligns with the current CLIP embedding, though some deviation is expected, as the MS-COCO validation set is only an approximation of the full training set. For completeness, the Appendix includes the same experiment with the text ellipsoid shifted instead of the image, showing consistent behavior. To conclude:

> **Property 6:** CLIP's loss is optimized for non-origin-centered ellipsoids, balancing alignment and uniformity for both correct and misclassified instances.

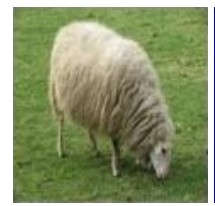
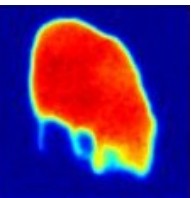

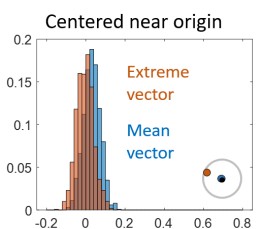
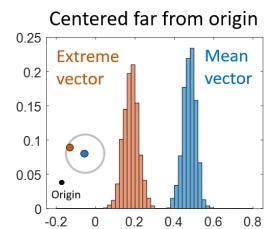

*Figure 7.* Top: Example of segmentation score blur (right), common in semantic segmentation, as object-membership uncertainty increases. Bottom: Similarity histograms of normally distributed samples for the mean vector (blue) and the furthest vector from the mean ("extreme", orange). Results are shown for a sphere centered near the origin (left) and one centered far from the origin (right). In contrastive learning, blur can be controlled by adjusting the sphere's offset. Embedding vectors closer to the center induces blur, while positioning them farther away sharpens the response.

## 6. False negatives and conformity

We demonstrate how the embedding geometry discussed earlier provides advantages in handling false negatives. Additionally, we introduce the concept of *conformity*, which plays a major role in forming the latent space distribution. A well-known challenge in contrastive learning is the presence of false negatives—pairs with similar meanings that are not dedicated pairs. Such samples should not be embedded far apart, as they fail to represent true negatives effectively. This issue arises in both single- and multi-modality settings and has been addressed by proposing new training procedures or alternative contrastive losses (Byun et al., 2024; Chuang et al., 2020). In CLIP, training uses a contrastive loss that does not explicitly address false negatives. However, we argue that this issue is partially mitigated by the embedding geometry. In classification and segmentation tasks, uncertainty typically results in softer predictions that reflect lower class membership probabilities. For example, Fig. 7 (top) illustrates a segmentation score where reduced confidence blurs the sheep's boundary, a phenomenon we term *semantic blur*. For contrastive networks, when false negatives are present, we expect lower confidence and a blurred response. On a high-dimensional sphere centered at the origin, such blurring is challenging, as small perturbations lead to large changes in cosine distance. We show that shifting the sphere away from the origin can effectively mitigate this issue. Concurrently, and closely related, Schrodi et al. (2024) discuss the relationship between entropy and

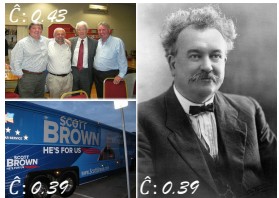
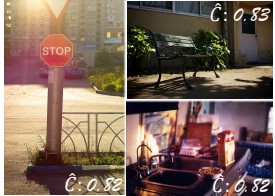

*Figure 8.* **High and low conformity of MS-COCO.** Low-conformity images often depict unique, distinguishable individuals or objects, whereas high-conformity images capture common scenes that could be found anywhere.

the modality gap.

**Blur through a non-origin centered sphere.** Fig. 7 (bottom) illustrates the difference between origin-centered and non-origin-centered spheres through an experiment. We draw 1000 random vectors $v^j \in \mathbb{R}^{512}$, where each element follows an independent Gaussian distribution with unit standard deviation. In the first experiment (Fig. 7, bottom left), the sphere is centered at the origin with an empirical mean $m$ close to zero. The blue histogram shows $\cos(m, v^j)$ for $j = 1, \ldots, 1000$. We then identify the furthest vector from $m$, $v^{\text{far}} = \arg\min \cos(m, v^j)$, and plot the histogram of $\cos(v^{\text{far}}, v^j)$ (orange), excluding $v^{\text{far}}$. In the second experiment (Fig. 7, bottom right), the sphere is centered at $(10, 5, 5, 0, 0, \ldots)$, modeling three dominant features with a mean distinctly far from zero. The same trial is repeated. The results highlight a significant difference: for an origin-centered sphere, the distributions of cosine similarity for the mean and the extreme vector are similar. In contrast, for a non-origin-centered sphere, the mean vector exhibits much higher average similarity. This allows the network to embed vectors with uncertainty closer to the mean, enabling *semantic blur*—reduced contrast in the response. This analysis, supporting a non-zero mean, leads to the following prediction:

> **Prediction 1:** Common themes, which occur more frequently in the training set, are expected to be embedded in closer proximity to the mean vector.

### 6.1. Conformity

To validate Prediction 1, we first formalize the term *common themes*, by defining a new notion, termed *conformity*.

**Definition 1** (Conformity). *Conformity of a vector $v^j$ within a set $S$ measures the expected value of the cosine similarity to this vector:*

$$C(v^j) = \mathop{\mathbb{E}}_{\substack{v^k \in S \\ j \neq k}} [\cos(v^j, v^k)], \tag{10}$$

*where for a given finite set $S$, the empirical mean is taken.*

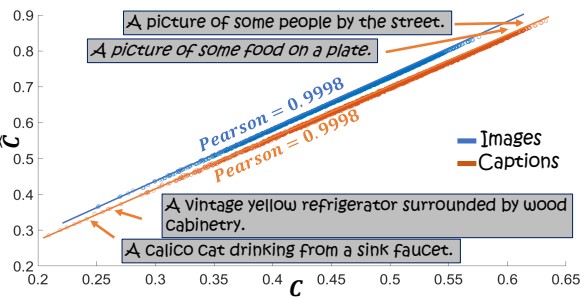

Figure 9. **Conformity.** Estimated conformity $\hat{C}$, Eq. 11, against conformity $C$, Eq. 13, on MS-COCO (Lin et al., 2014). The correlation is almost perfect. We can thus use the proposed estimated conformity reliably to quantify how common a sample is. More exotic captions have lower conformity (all examples are of eight words).

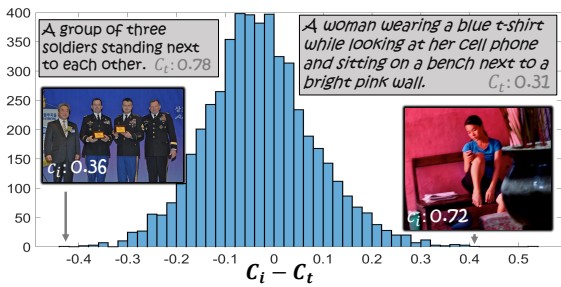

Figure 10. **Conformity Differences.** The conformity distributions of text and image modalities differ, as a common image may be described by a unique caption, and vice versa.

To provide more intuition, we present examples of high and low conformity from MS-COCO in Fig. 8, as well as on ImageNet-a (Hendrycks et al., 2021b) and ImageNet-R (Hendrycks et al., 2021a) in the Appendix. Following our prediction above, we propose a surrogate measure of conformity (which is much faster to compute). The estimation uses the following definition.

**Definition 2** (Estimated conformity)**.** *In contrastive learning embedding, for a given set of vectors $S$ with mean $m = \mathbb{E}_{v \in S}[v]$, the estimated conformity of $v^j \in S$ is:*

$$\hat{C}(v^j) = a \cdot \cos(m, v^j) + b, \qquad (11)$$

*where $a$ and $b$ are scalars determined by the embedding.*

In Appendix C1 we prove this correlation under the thin-shell assumption, and in Fig. 9, $C$ versus $\hat{C}$ are plotted for the entire MS-COCO set, for both image and text embeddings. A close to perfect correlation is obtained, with Pearson correlation of 0.9998 for both image and text where $a = 1.411$, $b = -0.008$ for text and $a = 1.461$, $b = -0.002$ for images, validating with close to perfect alignment with the rigorous mathematic derivation.

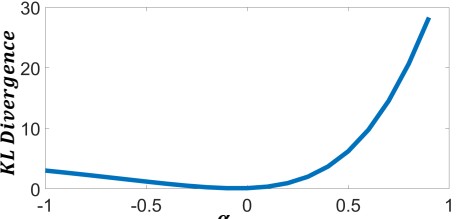

Figure 11. **Modality Gap matches conformity distributions.** The parameter $\alpha$ controls the embedding offset from the origin (as shown in Fig. 6). When $\alpha \approx 0$, i.e., the trained setting, image and text conformity distributions align well, with $KL_{\alpha=0} \approx 0.14$ indicating good distribution matching.

### 6.2. Modality gap assists in distribution matching

We now aim to provide a reason that can justify the presence of the well known *modality gap* (Liang et al., 2022). Our rationale for that phenomenon is as follows. The same incentive of having a mean not centered at the origin applies for both image and text modalities. However, in a single image-pair instance the uncertainty for each modality may differ (see Fig. 10). The same arguments as before promote uncertain instances to be near the mean and certain ones to be far from it. If both image and text of a pair are embedded at the same location - we may get contradicting requirements. Having separate embeddings for text and image allows to control the uncertainty of each instance for each modality. More generally, we would like to match the distribution of the conformity of both modalities. In Fig. 11 we show the KL-divergence of the conformity distribution as a function of $\alpha$, a parameter controlling the distance of the mean from the origin, as in Eq. 9, see illustration in Fig. 6. We show that the best distribution match is near $\alpha = 0$, i.e., in the current embedding of CLIP.

## 7. Applications

### 7.1. Conformity as a measure of expressiveness

We propose using conformity as a metric to assess generative method diversity. We measure conformity in images generated from MS-COCO captions by unCLIP (Ramesh et al., 2022) and Glide (Nichol et al., 2021), as shown in Fig. 13. Glide-generated images exhibit high conformity, indicating low detail and diversity, while unCLIP images are more varied and detailed. Both models, however, lack the diversity seen in real images. Similarly, we evaluate captioning methods by measuring conformity in captions generated by ClipCap (Mokady et al., 2021) and Caption Reward (Cho et al., 2022). ClipCap produces common captions, while Caption Reward generates diverse captions that even surpass human annotations.

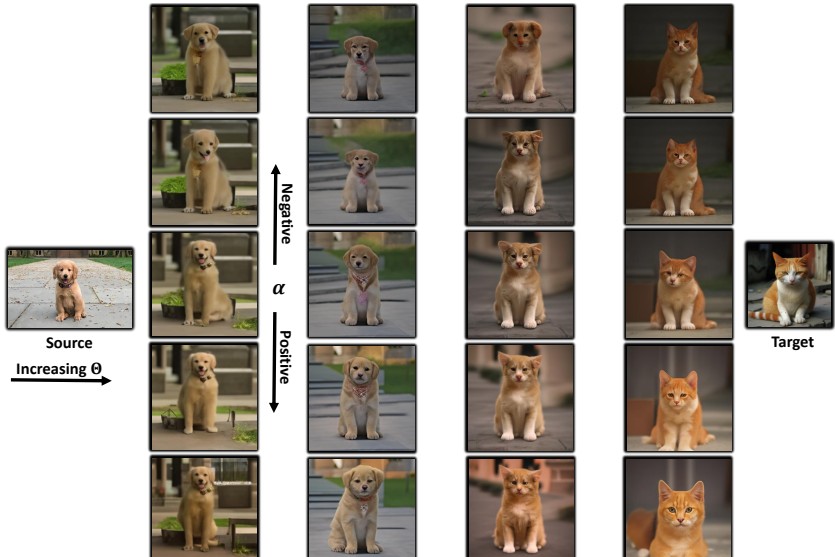

*Figure 12. **Vertical SLERP (vSLERP)** enables optimization-free, semantic editing. Interpolated images preserve the object with pose variations and roughly maintain backgrounds, with interpolation magnitude controlled by $\alpha$.*

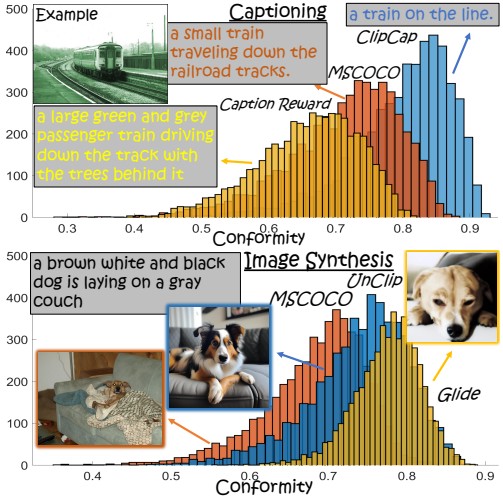

*Figure 13.* **Conformity analysis of captioning and image synthesis.** *Image Synthesis (top):* Glide generates more common images with less fine detail, while unCLIP creates more detailed images closer to natural distributions. *Captioning (bottom):* ClipCap produces more common captions, while Caption Reward generates more unique captions, even surpassing human annotations.

### 7.2. Unguided, training-free semantic generation

The unCLIP framework (Ramesh et al., 2022) introduces an image interpolation technique using spherical linear interpolation (SLERP) to transform a source image into a target image gradually. While this method produces visually appealing results, it often fails to preserve the same instance along interpolation, instead generating random instances.

In Fig. 12, we show images generated by an extension of SLERP, which we term as *vertical SLERP (vSLERP)*:

$$vSLERP(v^j, v^k, \theta_0, \alpha) = SLERP(v^j - \alpha m, v^k - \alpha m, \theta_0) + \alpha m$$
(12)

For brevity, $m_i$ and $v_i$ are referred to as $m$ and $v$. With a fixed $\Theta = \Theta_0$, adjusting $\alpha$ allows controlled manipulation of the same instance. This approach parallels real-image editing techniques; however, unlike methods relying on text inversion (Han et al., 2024; Gal et al., 2022; Mokady et al., 2023) or test-time optimization (Kawar et al., 2023), which are computationally heavy, *vSLERP* requires no training or optimization, thus, highly efficient.

## 8. Discussion and Conclusion

The paper examines the primary CLIP embedding, prior to projection onto the unit sphere, revealing that each modality forms a distinct, shifted ellipsoid with unique centers and radii. This geometry is the source of the modality gap and narrow cone phenomena (Liang et al., 2022; Schrodi et al., 2024; Afham et al., 2022), previously observed on the unit sphere embedding. We introduced *conformity*, a measure of similarity of an instance with an entire representative set. Our analysis shows that each modality exhibits a unique conformity distribution, with optimal alignment achieved when the ellipsoids are shifted from the origin. This provides a useful tool for assessing the diversity of captioning and image synthesis methods. Finally, we propose *vertical SLERP* (vSLERP), a training-free interpolation technique for specific object interpolation.

**Acknowledgements**

We would like to acknowledge support by the Israel Science Foundation (Grant 1472/23) and by the Ministry of Science and Technology (Grant No. 5074/22).

## Impact Statement

Our work advances machine learning by improving the geometrical understanding of CLIP's latent space. The findings may influence downstream tasks, though specific societal consequences do not need further emphasis here.

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

## A. Enlraged Visualizations

In Figure 15 and Figure 16, we provide the same visualizations as in the main paper, but enlraged, to enhance visibility. **CLIP of higher dimension.** We also show some results for CLIP with ViT-L/14 encoders, $n = 768$. In Figure 17 we show the distinct different statistics of image and text, mostly appearing in several pronounced features. Here as well, linear separation (100% classification accuracy) can be reached with only two features. In Figure 18 we show that the embedding can also be modeled as two separate thin shell ellipsoids for image and text.

## B. Statistical Analysis

We provide here the definitions of log concave distributions and isotropic random vectors, notions which are used in Section 4 of the main paper.

**Definition 3** (Log concave distribution). *A log concave distribution in $R^n$ has a density $p$ which admits, $\forall x, y \in R^n, \lambda \in [0, 1]$,*

$$p(\lambda x + (1 - \lambda)y) \geq p(x)^\lambda p(y)^{1-\lambda}.$$

The above definition is equivalent to stating that the logarithm of the density function is concave $\log p(\lambda x + (1 - \lambda)y) \geq \lambda \log p(x) + (1 - \lambda) \log p(y)$. Many well-known distributions admit this property, such as normal and multivariate normal distributions, exponential, Laplace, chi, Dirichlet, gamma and more.

**Definition 4** (Isotropic random vector). *A random vector $x \in R^n$ is isotropic if $\mathbb{E}[x] = 0$ and $\Sigma = I$, where $\Sigma$ is the covariance matrix of $x$ and $I$ is the identity matrix.*

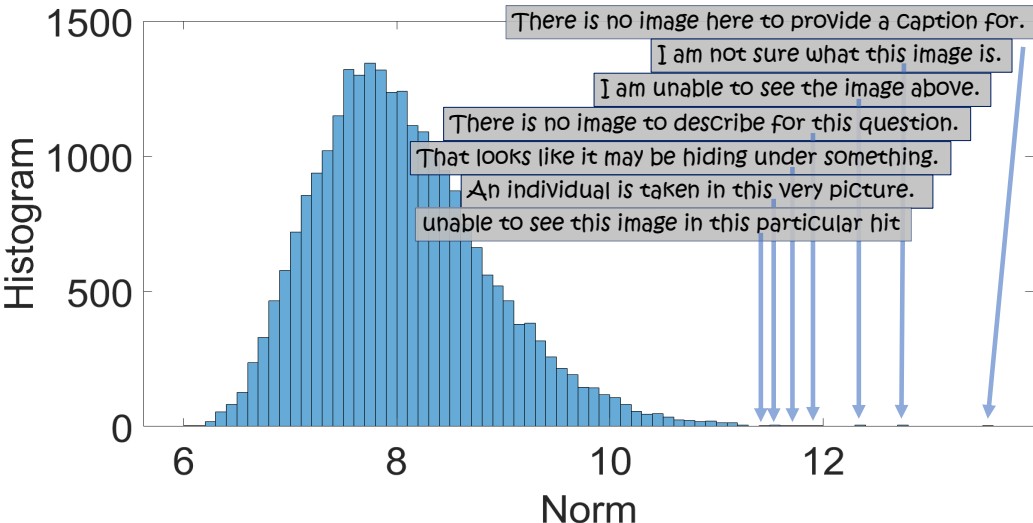

*Figure 14.* **Norm distribution.** While norm magnitudes are disregarded during training due to the normalization inherent in cosine similarity, they still capture meaningful semantic information.

We give below additional analysis related to applying a linear transformation that turns each ellipsoid into a sphere. This process is termed sphering or whitening. For lack of space, this part did not get into the main paper. However, we believe this analysis is of sufficient merit to be presented here.

## C. Additional Experiments and Visualizations

### C.1. Close relations between conformity and surrogate-conformity

We show below the validity of our conformity approximation under the thin-shell assumption.

**Proposition 1.** *Let $S = \{v^1, \ldots, v^N\}$ be a set of $N$ vectors in $\mathbb{R}^F$ exhibiting the* thin-shell phenomenon, *i.e.,*

$$\|v^i - \bar{v}\| \approx R \quad \text{for all } i,$$

where $\bar{v} = \frac{1}{N} \sum_{j=1}^{N} v^j$ *is the sample mean and we use the Euclidean norm* $\|v^i\|^2 = \sum_k (v_k^i)^2$. *Then, for any* $v^j \in S$, *the following approximation holds:*

$$\mathbb{E}_{v^j \in S}[\cos(v^i, v^j)] \approx A \cdot \cos(v^i, \bar{v}), \tag{13}$$

*where* $A \approx \frac{\sqrt{\mu_{norm}^2 + R^2}}{\mu_{norm}}$, $\mu_{norm} = \|\bar{v}\|$ *and the symbol* $\approx$ *represents the shell approximation (which becomes more accurate as the width of the shell decreases) and approximate orthogonality between a random vector and the mean vector.*

*Proof.* We start by expanding the left-hand side:

$$\mathbb{E}_{v^j \in S}[\cos(v^i, v^j)] = \frac{1}{N} \sum_{j=1}^{N} \frac{v^i \cdot v^j}{\|v^i\| \cdot \|v^j\|}.$$

Writing explicitly the inner-product we have:

$$\frac{1}{N \cdot \|v^i\|} \sum_{j=1}^{N} \frac{1}{\|v^j\|} \sum_{k=1}^{F} v_k^i v_k^j.$$

Now, consider the right-hand side of Equation (13):

$$\cos(v^i, \bar{v}) = \frac{v^i \cdot \bar{v}}{\|v^i\| \cdot \|\bar{v}\|} = \frac{1}{\|v^i\| \cdot \mu_{\text{norm}}} \sum_{k=1}^{F} v_k^i \left( \frac{1}{N} \sum_{j=1}^{N} v_k^j \right) = \frac{1}{N \cdot \|v^i\| \cdot \mu_{\text{norm}}} \sum_{j=1}^{N} \sum_{k=1}^{F} v_k^i v_k^j.$$

Observe that the only difference between the two expressions lies in the difference between $\mu_{\text{norm}}$ and $\|v^j\|$. We show below that under the thin-shell assumption $\|v^j\| \approx \sqrt{R^2 + \mu_{\text{norm}}^2}$.

Let us define by $z^j$ the difference vector between a vector $v^j$ and the mean vector $\bar{v}$, that is $z^j = v^j - \bar{v}$. Then,

$$\|v^j\|^2 = \|z^j + \bar{v}\|^2 = \|z^j\|^2 + 2z^j \cdot \bar{v} + \|\bar{v}\|^2.$$

In high dimensions, the inner product $z^j \cdot \bar{v}$ is small due to approximate orthogonality, so:

$$\|v^j\|^2 \approx \|z^j\|^2 + \mu_{\text{norm}}^2 \approx R^2 + \mu_{\text{norm}}^2.$$

Taking square roots:

$$\|v^j\| \approx \sqrt{R^2 + \mu_{\text{norm}}^2}.$$

Thus, the scalar factor $A$ in Equation (13) is given by:

$$A = \frac{\mu_{\text{norm}}}{\|v^j\|} \approx \frac{\mu_{\text{norm}}}{\sqrt{R^2 + \mu_{\text{norm}}^2}}.$$

$\square$

Empirically we know for Vit-B/32 that $\mu_{norm} = 7.587$ and $R \approx 7.59$, thus the mathematical derivation state that $A^{-1} = \frac{\sqrt{7.59^2 + 7.587^2}}{7.587} = 1.414$ For images and $A^{-1} = \frac{\sqrt{5.59^2 + 5.75^2}}{5.75} = 1.4$, very close to the empirical observations (note that the correlation is reversed in the main paper).

## C.2. Conformity

High- and Low-Conformity Images. We provide additional visualizations of high- and low-conformity images across various datasets. Figure 19 illustrates examples of sketches from ImageNet-R, while Figure 20 showcases examples from ImageNet-A. Both datasets contain out-of-distribution examples: ImageNet-A emphasizes natural adversarial images, while ImageNet-R features renditions of objects, such as origami or sketches.

From these visualizations, we observe that high-conformity images tend to contain less information. Sketches are simpler, and natural images often feature large uniform backgrounds or repetitive structures. In contrast, low-conformity images frequently include substantial text, while natural images exhibit collages of objects with unique or diverse colors.

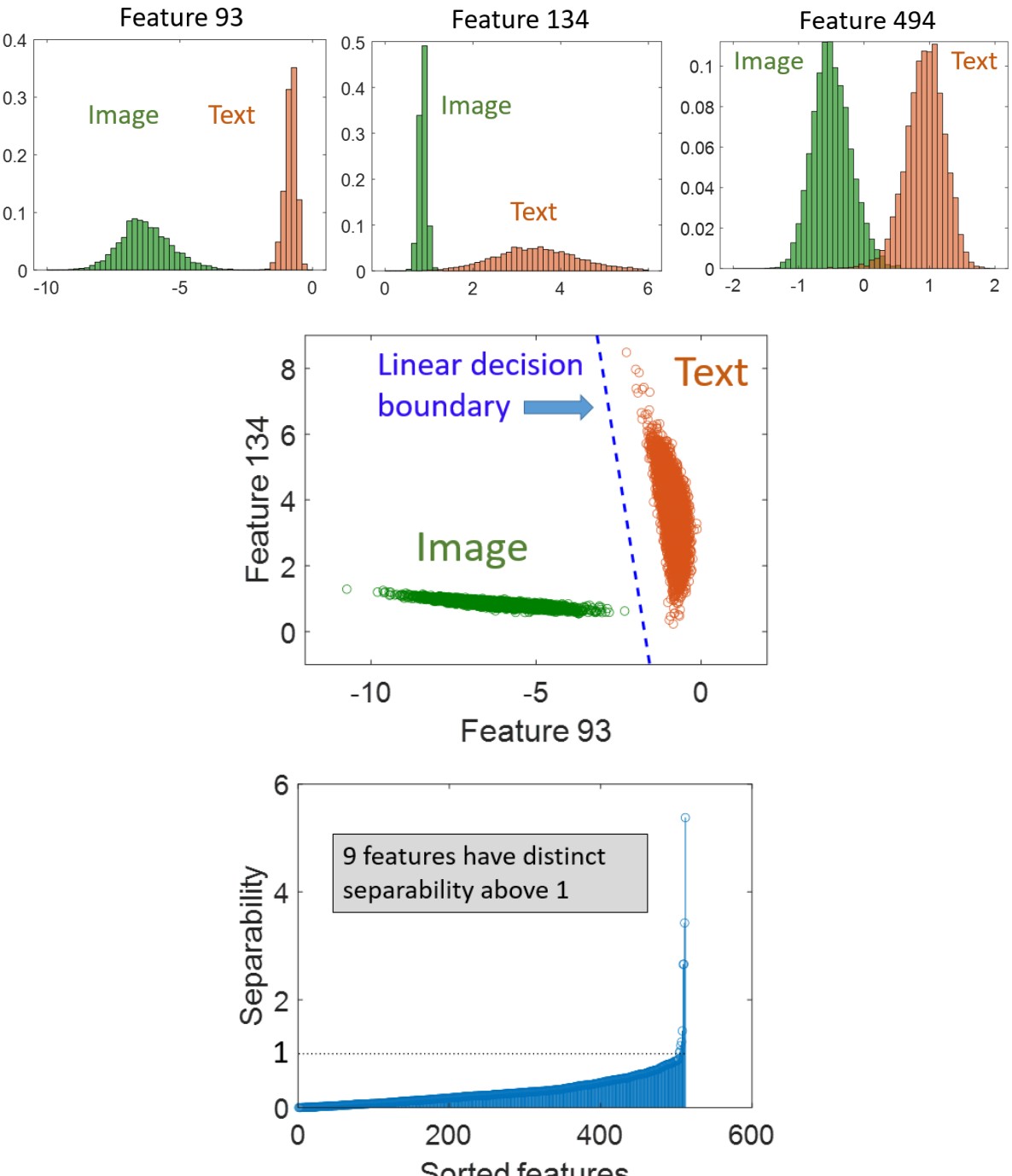

*Figure 15.* **Enlarged plots from Section 4.**

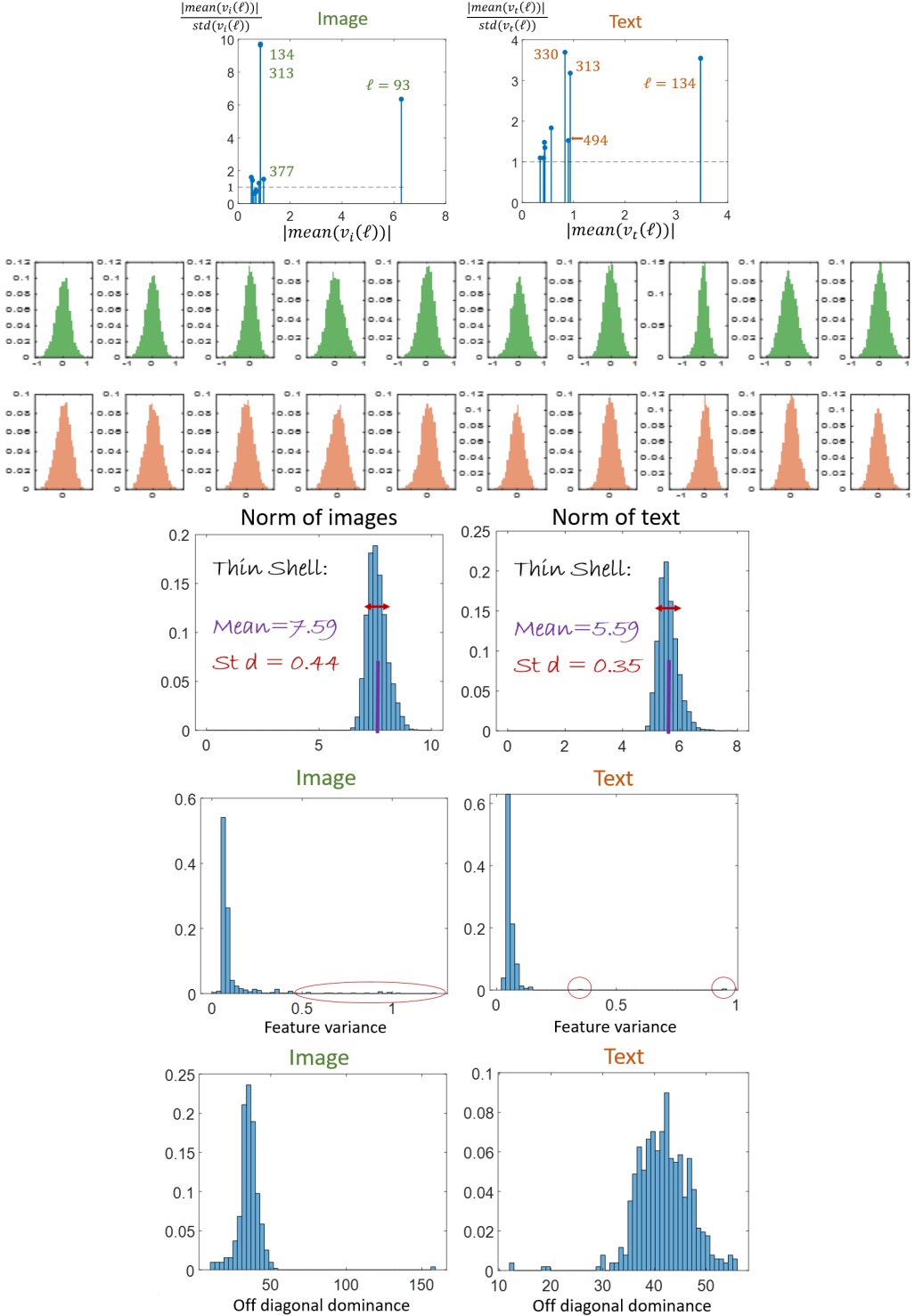

*Figure 16.* **Enlarged plots from Section 4.**

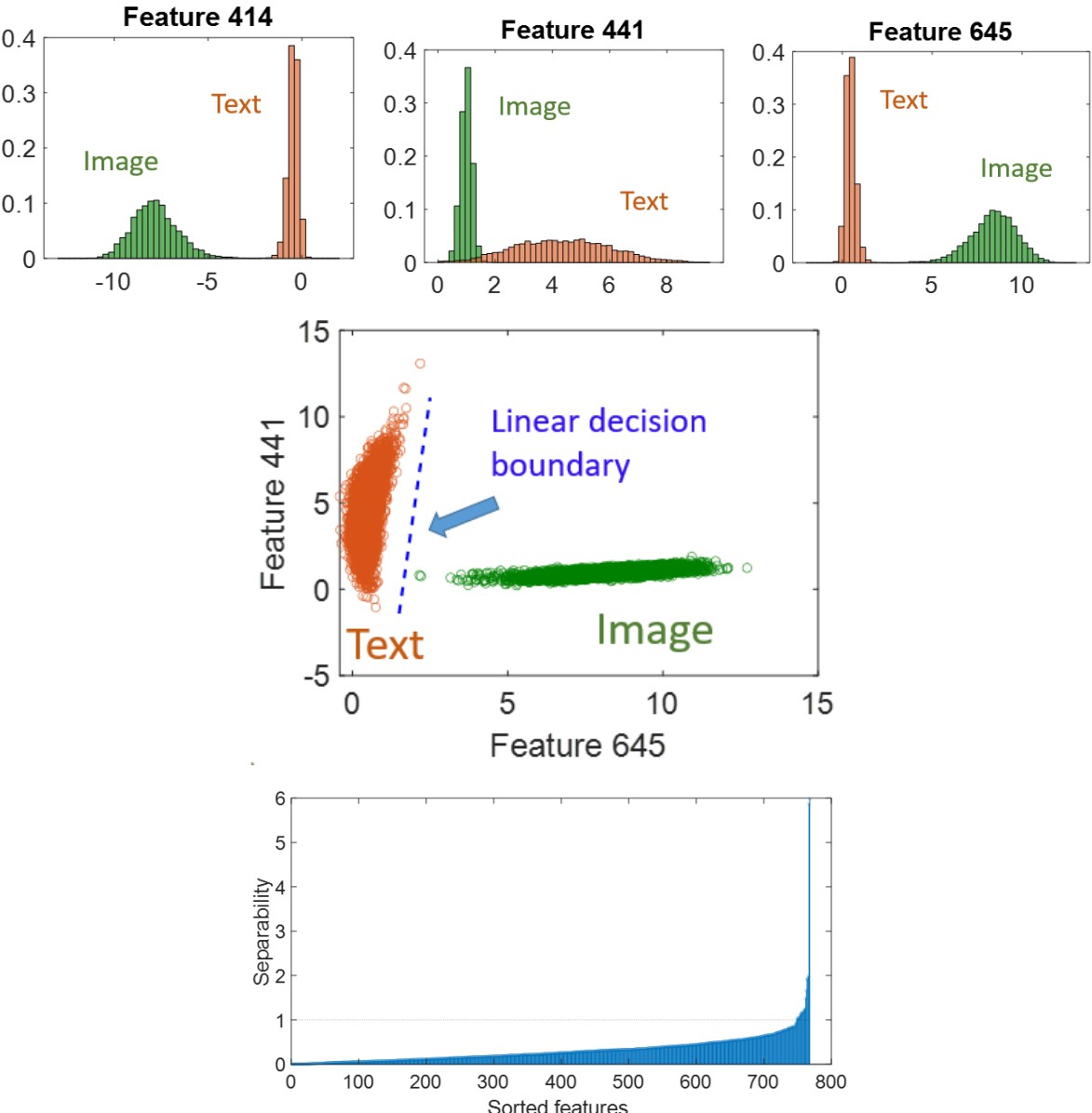

*Figure 17.* **Enlarged plots for CLIP embedding of** $n = 768$**.** There are dominant features with clearly different distribution between image and text. Both modalities can be separated (with perfect accuracy) by a linear SVM classifier based on only 2 features. With respect to separability (bottom), there are 20 features with value above 1.

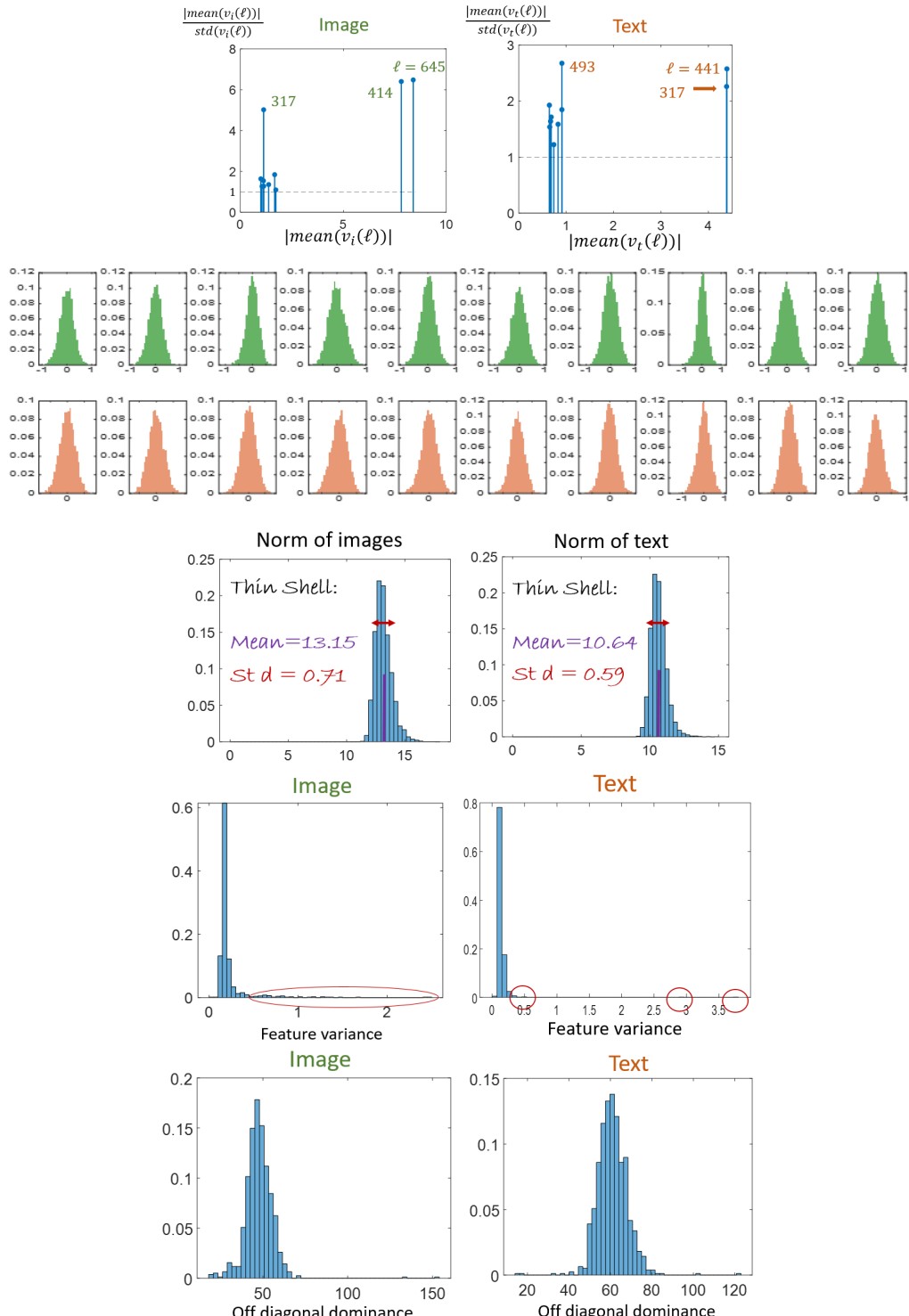

*Figure 18.* **CLIP** $n = 768$**, thin shell phenomenon.** We can observe similar geometry (as in the case of $n = 512$) of two tilted ellipsoids, one for each modality, not centered at the origin.

## C.3. Reaffirming loss and conformity matching experiments

We revisit the loss experiment presented in Fig. 6 of the main paper and the conformity matching experiment shown in Fig. 11. To further validate our findings, we conduct these experiments under two alternative settings.

First, we shift the text ellipsoid instead of the image ellipsoid, applying the following transformation:

$$v_t^{j'} = v_t^j - \alpha \cdot m_t \quad \forall j \in M, \tag{14}$$

where the values of $v_i$ remain unchanged. The results of this experiment are presented in Figure 21.

In the second setting, we align both the image and text ellipsoids at the origin by applying the following transformations:

$$v_t^{j'} = v_t^j - \alpha \cdot m_t, \quad v_i^{j'} = v_i^j - \alpha \cdot m_i \quad \forall j \in M. \tag{15}$$

Here, for $\alpha = 0$, the ellipsoids remain in their optimal positions after training, while for $\alpha = 1$, both ellipsoids are shifted to the origin as in Figure 22.

Both experiments reaffirm that the current positioning of the ellipsoids yields optimal results in terms of loss and conformity matching. These findings further support our claims across different alignment scenarios.

## C.4. vSLERP

Here, we provide additional examples of vSLERP, shown in Figure 23 and Figure 24. As discussed in the main paper, the standard SLERP process typically generates interpolated images representing different objects or individuals. In contrast, our proposed vSLERP method produces diverse variations of the same object.

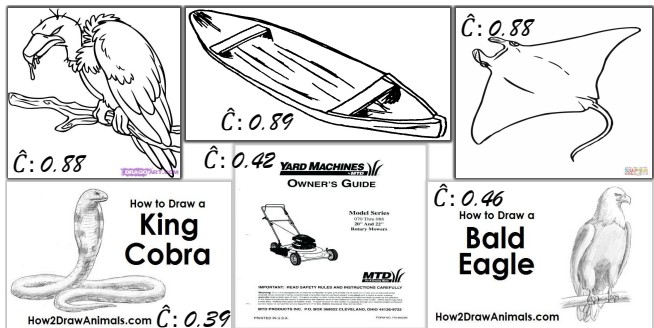

*Figure 19.* **High and low conformity of sketches from ImageNet-R.** Images with high conformity tend to be simpler and cleaner, while low-conformity images often feature complex details covered by large portions of text descriptions.

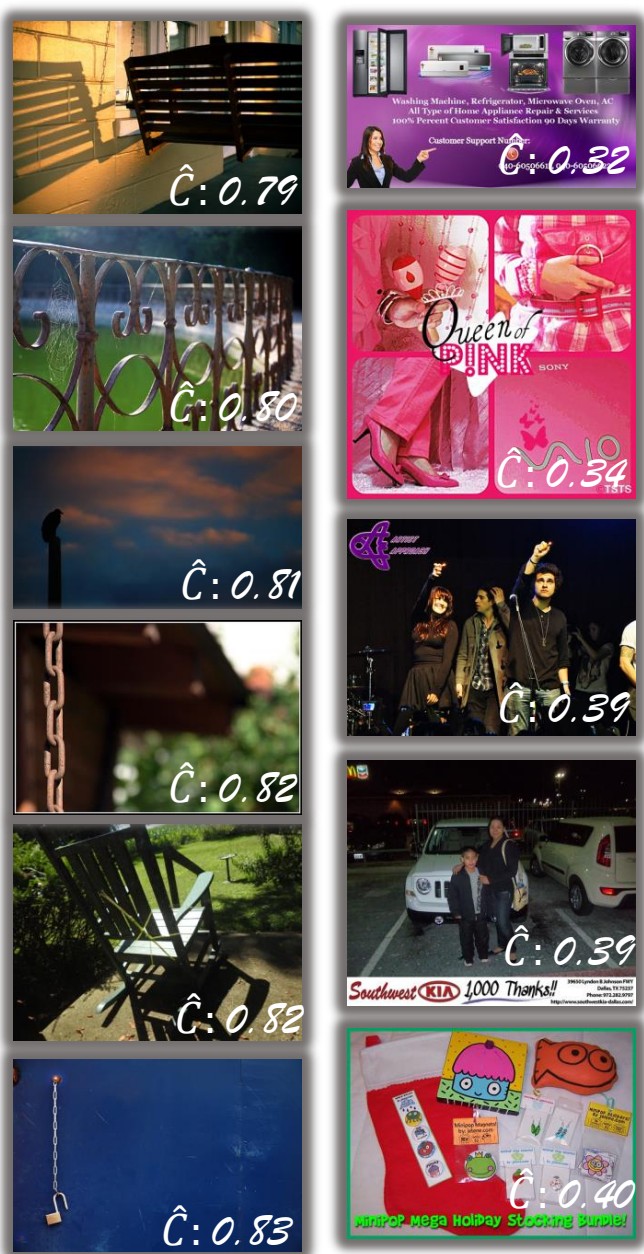

*Figure 20.* **Conformity on ImageNet-a.** It is possible that high conformity images are with more unique colors, perhaps contains people or text, whereas low conformity images tends to contain low amount of information.

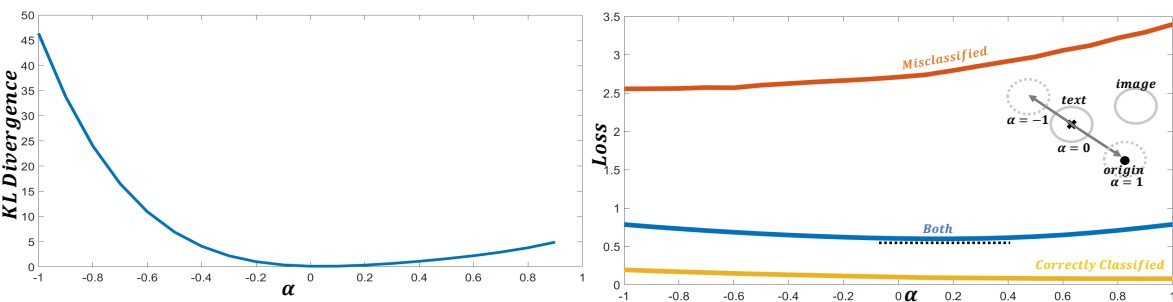

*Figure 21.* **Shifting text ellipsoid only.** Conformity distribution matching and loss experiments when shifting text ellipsoid only as in Equation (14)

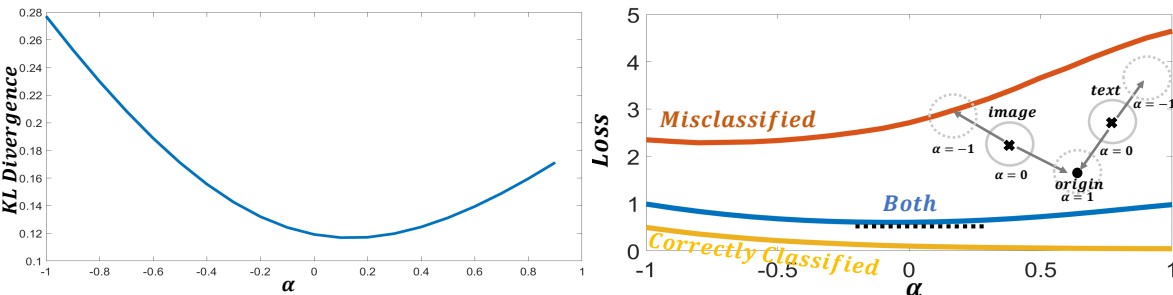

*Figure 22.* **Shifting both ellipsoids.** Conformity distribution matching and loss experiments when shifting both text and image ellipsoids as in Equation (15).

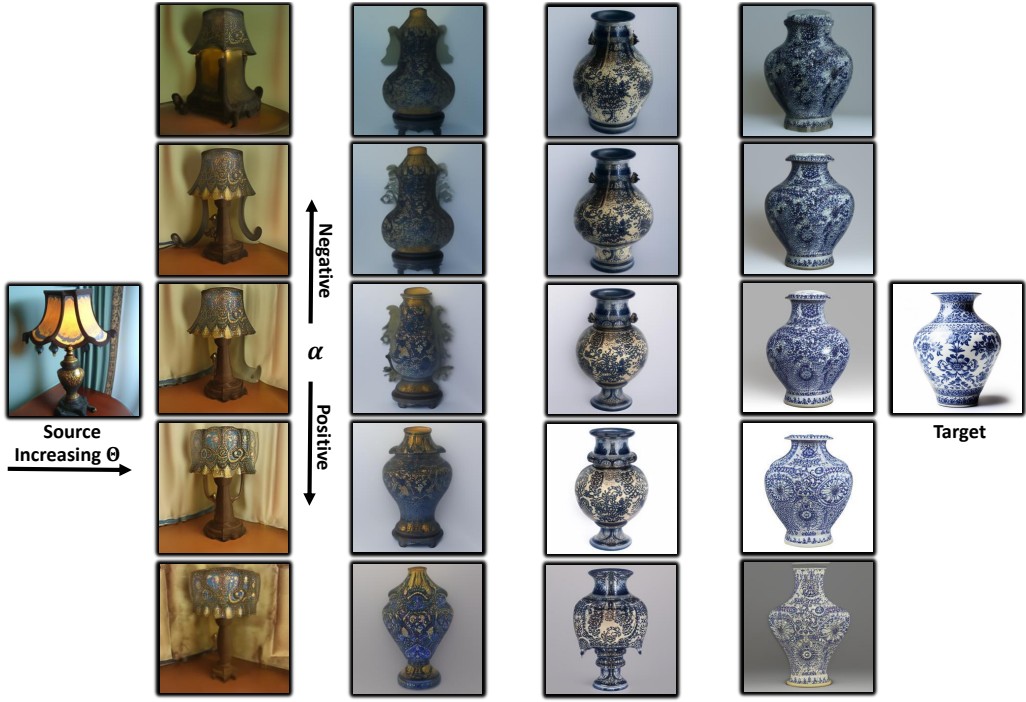

*Figure 23.* **vSLERP lamp to vase.**

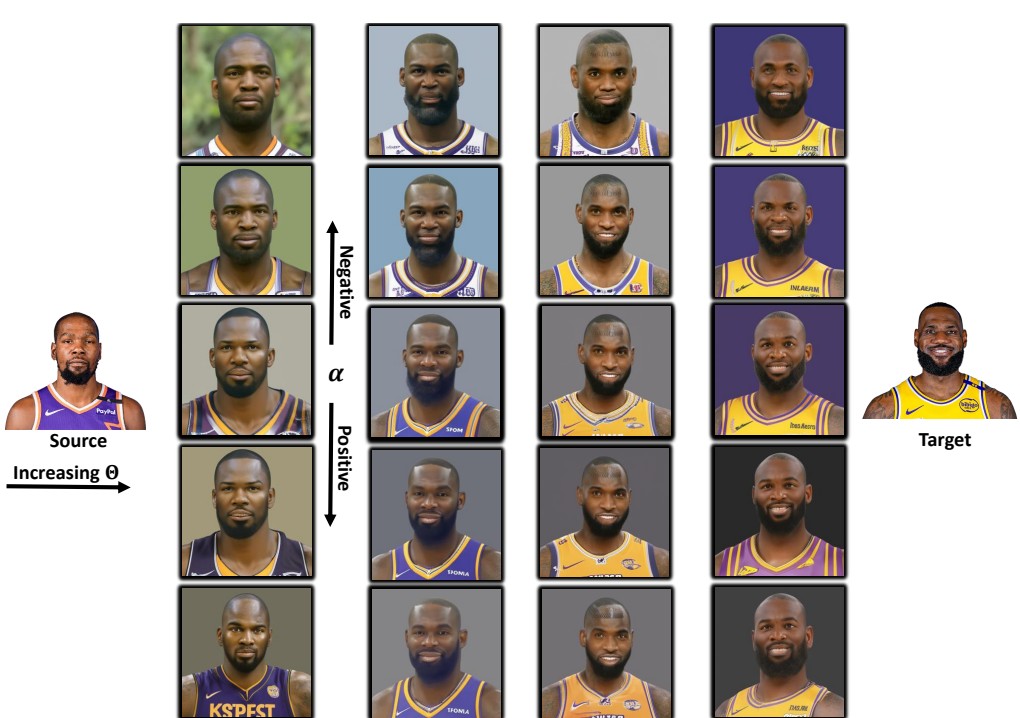

*Figure 24.* **vSLERP Kevin Durant to Lebron James.**

