# OpenReview forum: "The Double-Ellipsoid Geometry of CLIP"
_ICML.cc/2025/Conference — ICML 2025 poster_

### Official Review · Reviewer_qxx2 · 2025-03-11

**Overall Recommendation:** 4

**Summary:**

The work analyses the geometric properties of CLIP embeddings.  It builds upon previous work that studied the modality gap in embeddings.  The main finding is that image and text embeddings both live within separate ellipsoid thin shells in high dimensional embedding space.  The authors demonstrate that this particular geometric configuration arises from the contrastive loss function and noise in the training dataset, i.e. patches with similar meaning that are not dedicated pairs (and thus used as "false" negatives in the loss).  The paper introduces a measure of "conformity", indicating that embeddings of common themes (in image or text space) should be close to the mean embedding vector.  Based on this definition, the authors provide a geometric justification for the modality gap, i.e. the present non-overlapping distributions for text and image embeddings minimize the KL divergence of the conformity distributions.

**Claims And Evidence:**

Numerous claims are made in the manuscript, which are all supported well.  I particularly enjoyed the simple but effective geometric interpretation of the results, e.g. Fig 6 or 11.

**Essential References Not Discussed:**

N/A

**Experimental Designs Or Analyses:**

See other comments. The paper itself is a detailed problem analysis.

**Methods And Evaluation Criteria:**

The geometric analysis is based on the theory of random vectors in high dimensional spaces and thus well grounded.  Experiments are conducted on the MS-COCO dataset which has 328k images accompanied by natural language descriptions and thus seems adequate for the task.

**Other Comments Or Suggestions:**

N/A

**Other Strengths And Weaknesses:**

+ Well written paper. The analysis is easy to follow.

- The applications of Sec. 7 are only exemplary and further usage of the introduced conformity measure remains unclear.

**Questions For Authors:**

N/A

**Relation To Broader Scientific Literature:**

Yes.  The paper presents interesting findings, which are relevant for anyone using a contrastive loss across modalities in ML.

**Theoretical Claims:**

Claims are argued for by geometric interpretations and experiments, which appear correct.  The definition of conformity Eq. 10 is plausible The validity of its approximation Eq. 11 demonstrated by data, Fig. 9

---

> ### Author Rebuttal · Authors · 2025-03-30
>
> Thank you for your positive feedback. The applications section is indeed small since we devoted our paper to the geometric and statistical analysis, which we found the most important to delve in, explain and support experimentally. See additional comments embedded in our answers to the other two reviewers, which we do not want to repeat here.
>
> The applicative examples are preliminary, still we found them exciting and useful. Studying these applications more thoroughly and suggesting additional ones would require an in-depth analysis which we believe would better fit a dedicated paper. Thanks again for your feedback.

---

### Official Review · Reviewer_UMeX · 2025-03-13

**Overall Recommendation:** 1

**Summary:**

This paper investigates the geometric properties of the CLIP embedding space, proposing that image and text modalities form independent double-ellipsoid structures displaced from the origin. The authors argue that this structure improves the performance of contrastive learning and provides explanations for previously known phenomena such as the modality gap and narrow cone effect.

**Claims And Evidence:**

The main claim of the paper is that the CLIP embedding space exhibits a double-ellipsoid geometry rather than the simpler hyperspherical structure typically assumed in prior literature. However, this claim suffers from significant shortcomings:
- The key finding of the double-ellipsoid geometry largely reiterates existing results from prior studies, notably "Mind the Gap" (Liang et al., 2022), which has already thoroughly explored the modality gap and thin-shell phenomena. Thus, the contribution of this work does not clearly differentiate itself from earlier research, casting substantial doubt on the novelty and significance of the proposed ellipsoidal geometry.
- The "Conformity" concept introduced by the authors shows an extremely high Pearson correlation (0.9998) with the cosine similarity to the mean vector, questioning its practical novelty. This implies that the Conformity metric essentially duplicates existing cosine similarity measures and lacks a compelling justification for its introduction as a distinct concept.

**Essential References Not Discussed:**

"Mind the Gap" (Liang et al., 2022) would be one of the essential references. This paper already provides a comprehensive visual and analytical examination of the modality gap phenomenon in CLIP, making it directly relevant and necessary to cite given the main claims of this study.

**Experimental Designs Or Analyses:**

- The claim that the double-ellipsoid structure alleviates the False Negative problem in contrastive learning is not convincingly supported by empirical evidence. The authors do not disentangle the independent effects of the ellipsoidal structure itself from the impact of displacement from the origin, leaving the exact reason for any observed improvements unclear.

**Methods And Evaluation Criteria:**

- The experimental analyses conducted in this study are limited exclusively to a few samples in the MS-COCO dataset without any training or new empirical results.
- The proposed vertical SLERP (vSLERP) method lacks sufficient empirical evaluation. The authors fail to provide clear quantitative evidence of improvement over the traditional SLERP method or adequately explain theoretically why vSLERP better leverages the geometric properties of CLIP embeddings.

**Other Comments Or Suggestions:**

see the above

**Other Strengths And Weaknesses:**

see the above

**Questions For Authors:**

see the above

**Relation To Broader Scientific Literature:**

- Efforts to understand CLIP embeddings were widely explored before, however, I do not agree that this paper brings new things.

**Theoretical Claims:**

I have no issues here.

---

> ### Author Rebuttal · Authors · 2025-03-30
>
> - Limited novelty: as other reviewers pointed out, we study well known phenomena using a different lens, i.e. a geometric perspective. As far as we know, we are the first to analyze the raw features prior to the normalization phase. The normalized features are forming a unit hypersphere by definition, overshadowing the complicated double-ellipsoid structure lost in the projection stage. Thus, since no other paper investigate the raw features, we are not familiar with thin shell explanation prior to us (including in "Mind the Gap" where the words "thin" as well as "shell" are absent in the entire paper). Moreover, the work shows novel geometric findings which capture the popularity of concepts, referred by us as “conformity”.
>
> - Conformity essentially yield a scalar per feature correlated with an image/caption measuring its popularity (its average similarity with other concepts drawn at random). Unlike cosine similarity, which yields a similarity measure between two instances (image and/or text) as input, conformity has a single input. We show conformity is almost perfectly aligned with cosine similarity to the mean of the ellipsoid of the respective modality. This finding has two significant consequences: From an applied perspective, it is fast and easy to compute (only one cosine distance computation with a given vector). From a contrastive-learning perspective, we highlight an interesting novel phenomenon: frequent concepts are embedded closer to the modality mean. We also explain the rationale, by reducing the loss of false negatives, which are expected to occur more for frequent concepts.
>
> - Our primary goal is to better understand the existing embedding space of CLIP, therefore we refrained from training or changing it. The natural follow-up study could be to leverage this new knowledge to retrain CLIP in a better fashion in some sense. We share directions for better training with reviewer MmZu. Regarding vSLERP, the main contribution is the preservation of the object appearance, which poses a challenge in real image editing.
>
> - Supp. are provided. According to the ICML policy, it appears following the main paper and includes additional clarifications and visualizations. We kindly encourage the reviewer to examine it.
>
>
> - "Mind the Gap" (Liang 2022) is indeed a pioneering paper in this field, consequently it is cited 12 times! (on pages 1,2,3,4,7 and 8, in some pages multiple times). We believe we carefully gave it the appropriate credit.

---

> > ### Comment · Reviewer_UMeX · 2025-04-04
> >
> > Thank you for the rebuttal. However, it does not sufficiently address my initial concerns. In particular, the claim of novelty from analyzing raw features and the Conformity metric remains unconvincing. Additionally, the rebuttal does not adequately respond to my concerns about the limited experimental validation, which still relies on only a small subset of MS-COCO samples. Lastly, the visual evidence solely provided for vSLERP remains insufficient to demonstrate a clear practical advantage over the conventional SLERP method. I currently maintain my original score.

---

### Official Review · Reviewer_MmZu · 2025-03-16

**Overall Recommendation:** 4

**Summary:**

This paper investigates the geometry of the pre-normalized CLIP embedding space. The main finding is that image and text embeddings reside on linearly separable ellipsoid shells, which are not centered at the origin. This non-origin-centered, double-ellipsoid structure is proposed as a key factor in controlling uncertainty during contrastive learning, where more frequent concepts with higher uncertainty are embedded closer to the modality mean vector, a phenomenon the authors term "semantic blurring."

The paper introduces the concept of "conformity," defined as the expected cosine similarity of an instance to all other instances in a representative dataset. A significant result is the strong correlation (Pearson correlation: ~0.9998 on MS-COCO) between this conformity measure and the cosine similarity of an instance to the modality mean vector. Furthermore, the paper demonstrates that the modality gap observed in CLIP can be explained by the need to align the different conformity distributions of image and text, and that the current non-zero offset of the ellipsoids optimizes this alignment.

The paper claims to contribute the following conceptual ideas and findings: (1) revealing the double-ellipsoid geometry of CLIP embeddings, shifted from the origin; (2) analyzing the benefits of this geometry in controlling sharpness in contrastive learning and mitigating false negatives; (3) showing that frequent concepts benefit most from this geometry; (4) defining concept conformity and demonstrating its strong correlation with similarity to the mean vector; (5) highlighting the role of conformity in explaining the modality gap; and (6) proposing a new interpolation method, vertical SLERP (vSLERP), that leverages the identified geometric properties for improved semantic editing.

**Claims And Evidence:**

*   **Image and text reside on linearly separable ellipsoid shells, not centered at the origin:** The paper mentions statistical analysis of the MS-COCO validation set and Figure 1 as a sketch illustrating this geometry. Figures 4 and 5 are also referenced in relation to the thin-shell phenomenon and the non-uniform variance of features (suggesting an ellipsoid rather than a hypersphere). Figure 2 shows linear separability.
*   **Offset from the origin helps mitigate false negatives and control sharpness:** This is discussed in Section 4, and Figure 7 illustrates the concept of blur control through sphere offset.
*   **Frequent concepts are embedded closer to the mean vector (semantic blurring):** This is hypothesized in Section 4 and linked to the non-origin-centered geometry. The experiments confirm the better alignment of frequent concepts to the mean vector.
*   **Strong correlation between conformity and cosine similarity to the mean vector:** The paper explicitly states a Pearson correlation of 0.9998 on MS-COCO in Section 4, supported by Figure 9.
*   **Modality gap helps in aligning conformity distributions:** This is argued in Section 6.2 and visually supported by Figure 11, showing the KL-divergence of conformity distributions as a function of the mean offset.
*   **vSLERP leverages CLIP's latent space geometry for semantic editing:** Figure 12, Figure 23, and Figure 24 provide visual examples of the vSLERP method in action.
*   **Conformity as a measure of expressiveness:** Section 7.1 proposes this, supported by conformity measurements on generated images (Figure 13). Lower conformity means diversity

Need more evidence:

*   **The extent to which the embedding geometry "explains" the modality gap and narrow cone effect:** While the paper links these phenomena to the identified geometry, the depth of this explanation and whether it's fully convincing might require more detailed theoretical grounding.
*   **The "optimality" claimed for various aspects of CLIP's geometry:** The term "optimal" can be strong. The evidence would need to clearly demonstrate that the observed structure indeed leads to the best possible performance in relevant aspects (e.g., loss, alignment, uniformity, conformity matching) compared to alternative geometries.

**Essential References Not Discussed:**

N/A

**Experimental Designs Or Analyses:**

Sound:

- Analyzing Feature Statistics (Figure 2, 4, 5, 15, 16, 17, 18): check

- Linear Separability Analysis (Figure 2, 15, 17): check

- Thin Shell Phenomenon Analysis (Figure 4, 16, 18): check

- Conformity Analysis (Figure 9, 10): check

- Conformity as a Measure of Expressiveness (Figure 13): check

- Vertical SLERP (vSLERP) Demonstration (Figure 12, 23, 24): check

Minor Questionable:

- Though I believe there may not be much difference in conclusion, I am still wondering how MS-COCO training data differs.
- Will different image/text backbone matter? Or only the contrastive training nature matters?

**Methods And Evaluation Criteria:**

Methods:

The paper is theoretical analysis for the CLIP geometry. The major proposed method is Conformity and Estimated Conformity.
Conformity defines a quantitative measure for how common or unique an embedding is.
Estimated Conformity is based on cosine similarity to the mean vector proposed a computationally efficient surrogate for conformity. It makes practical sense for large datasets and real-world applications. The strong correlation demonstrated with the actual conformity measure supports its validity.


Evaluation:

- Using a standard and widely used image-text dataset like MS-COCO for statistical analysis of CLIP embeddings is appropriate for understanding general properties.
- Using KL-divergence to quantify the difference between the conformity distributions of image and text modalities is a standard and meaningful metric for comparing probability distributions. This makes sense in the context of analyzing the modality gap.
- The visual examples provided for vSLERP (Figure 12, 23, 24) are impressive to demonstrate its potential for preserving object identity during interpolation, which is a key challenge in semantic editing.
- Using the introduced "conformity" metric to assess the diversity of generated images and diversity of the generated captions makes more sense

**Other Comments Or Suggestions:**

N/A

**Other Strengths And Weaknesses:**

Originality:
- Modality gap is already discovered, but this paper added a geometric explanation
- The introduction of "conformity" as a measure of how common or unique an embedding is within a dataset is an original concept.
- Proposed applications: 1. measure of diversity 2. vSLERP are very impressive to make CLIP a better tool for wider community.

**Questions For Authors:**

Open questions:

- How can the analysis in this paper help with
1. a more efficient CLIP training?
2. a higher quality CLIP training?

- Can we potentially offset the ellipsoid to mitigate the modality gap to make the similarity between different modalitiy-pairs comparable?
e.g.
Image(Dog A), Image(Dog B) has cosine distance a
Image(Dog A), Text(Dog B) has cosine distance b
Due to the modality gap, cosine distance a !=b. and it is meaningless to compare a and b due to different modality. What if I want to make them comparable?

**Relation To Broader Scientific Literature:**

- The modality gap, where embeddings from different modalities (image and text) are separated, and the narrow cone effect, where features occupy a limited angular space, have been identified and studied in CLIP in previous works. The paper offers a geometric explanation for these phenomena based on the identified non-origin-centered ellipsoid shells for each modality.

- The introduction of "conformity" as a measure of expressiveness and diversity in generative models (like unCLIP and Glide) is a novel contribution. This helps to evaluate the diversity of generated content which is crucial for generative models.

- The issue of false negatives (semantically similar pairs treated as negatives) is a recognized challenge in contrastive learning. The paper suggests that the non-origin-centered ellipsoid geometry inherently helps in handling false negatives by allowing for "semantic blurring", where uncertain instances are embedded closer to the mean. This offers a different perspective – that the embedding structure itself plays a role in managing this issue, rather than solely relying on training procedures or loss modifications.

**Theoretical Claims:**

The mathematical definitions are sound. theoretical claims were derived on the basis of analysis and observation. Thus no proofs to check.

---

> ### Author Rebuttal · Authors · 2025-03-30
>
> Thank you for your positive feedback, intriguing comments and thought-provoking questions.
>
> - The geometry explains the modality gap and the narrow cone effect:
> We agree that the geometry does not fully explain the reasons to all observed phenomena. We have a strong evidence that the offset of the ellipsoids is alleviating the impact of false negatives. There have been several proposed efforts in the literature to mitigate the false negatives in contrastive learning. However, to the best of our knowledge, we are the first to make the link between false negatives and the geometry CLIP converged to.  Part of our message is that the geometry analysis should be performed in the native raw embedding, not on the unit sphere, which reduces information. Hence, “modality gap” is actually “linearly separable ellipsoids” and “narrow cone effect” is better understood as “non-origin-centered”. We believe our findings can facilitate future research on contrastive learning and are thus of interest to the community.
>
> - Optimality: All of the references to “optimality” are related to the losses mentioned in Figs 6 and 11. By “optimal” we mean that the value of alpha is attained at the loss minimum. We agree that it holds only under restricted cases, as explained in the paper, where not all possible combinations of geometry are examined (hence we cannot claim global optimality, with respect to the tested variables). This should indeed be clarified and toned down, as we intend to do in the final version.
>
> - Backbone/training procedure: we validate our findings on ViT-L as well (on top of ViT-B in the main paper), thus we feel confident that architecture size (number of layers/heads or even feature size) is probably not playing a significant role in forming the latent geometry. Further examinations on additional architectures or larger datasets are important, we plan this for future study. Since CLIP is a fundamental backbone of many vision and text algorithms today, we believe knowledge on its geometry is nevertheless of high significance.
>
> Questions:
>
> Leveraging geometric knowledge to enhance CLIP training efficiency and/or quality: possible direction could be to force geometric constraints during training, possibly by centralizing both ellipsoids as you suggested. In Fig. 22 (Supp.) we show the loss values when both ellipsoids are shifted to the origin (alpha=1). As can be seen, although correctly classified instances yield lower loss, the combination with the misclassified increase the loss. The reason is implicitly shown in Fig. 7. The average cosine similarity for non-centered ellipsoid is approx. 0.2 (highly likely that misclassified samples will lay there), whereas for centered ellipsoid it is around zero. This affects both the classified and misclassified, increasing the loss for the latter. Alternatively, one may propose an alignment which does not necessarily shift both cases to the origin. By doing so we can decouple the modality gap from the narrow cone effect, hopefully mitigating at least one of them. We are eager to further study this direction from a geometric perspective. We hope our clarifications answer your concerns and if so we would be delighted if  the final paper rank can be upgraded.

---

> > ### Comment · Reviewer_MmZu · 2025-04-09
> >
> > Thanks for the clear rebuttal, I found the work is very useful for the multimodal representation learning.
> > Another potential direction can be beyond unimodal, towards modality combinations, like Image(black dress) + Text(red) -> Image(red dress). changing the rating to accept

---

### Decision · Program_Chairs · 2025-05-01

**Decision:**

Accept (poster)

**Comment:**

This paper analyzes the geometry of CLIP embeddings, showing that image and text embeddings lie on linearly separable ellipsoid shells. This paper introduces conformity to measure embedding similarity and explains how this structure manages uncertainty and aligns modality distributions. This paper offers an analysis of the modality gap and contrastive learning dynamics.

This paper has mixed opinions (2 Accept and 1 Reject). The AC and the reviewers had discussions, but their opinions didn't reach a consensus. More specifically, Reviewer UMeX pointed out three issues:

1. The conformity metric and analyzing raw features are not novel
2. Experiments are limited to the small subset of MS-COCO
3. The visual evidence is insufficient to demonstrate a clear practical advantage of the proposed method.

In terms of novelty, I think that the result of this paper is sufficiently novel compared to "Mind the Gap" (Liang et al., 2022). Although the "thin-shell phenomena" itself was revealed in Mind the Gap paper, I believe that the theoretical approach of this paper is different from the Mind the Gap paper (despite that the claim itself is not novel). I asked Reviewer UMeX to provide more evidence to support the "lack of novelty" comment, but I couldn't get a response from the reviewer. As far as the AC knows, the theoretical analysis of this paper (specifically, linking the false negative problem and conformity) is a sufficiently novel result.

Reviewer MmZu also has the same perspective on point 2 but stands an opposite site to point 3. -- Reviewer MmZu clarified that the experimental results are "good to have" but not critical. Reviewer qxx2 also clarified that the current experimental validation is limited, but the reviewer clarified that they think this work as a detailed problem analysis rather than the presentation of a solution to a certain problem, hence the evaluation is not a critical issue. I slightly lean to the other reviewers' opinions, but lack of the empirical validation could be a valid argument.

For the third issue, I think Reviewer UMeX's opinion is sound.

Overall, I think this paper is a borderline paper where the advantages of this paper (providing a new lens of VL embeddings) slightly outweigh its disadvantages (insufficient empirical results). I recommend "Weak Accept" for this paper; I don't think this paper is strong enough to be accepted, but if there is a room in the program, I am willing to recommend Accept.